# Association of insularity and body condition to cloacal bacteria prevalence in a small shorebird

José O. Valdebenito[1]*, Josué Martínez-de la Puente[2,3], Macarena Castro[4], Alejandro Pérez-Hurtado[4], Gustavo Tejera[5], Tamás Székely[1,6], Naerhulan Halimubieke[1], Julia Schroeder[7], Jordi Figuerola[2,3]

1 Milner Centre for Evolution, University of Bath, Bath, United Kingdom, 2 Department of Wetland Ecology, Estación Biológica de Doñana (EBD-CSIC), Seville, Spain, 3 CIBER Epidemiología y Salud Pública (CIBERESP), Seville, Spain, 4 Instituto Universitario de Investigación Marina, Facultad de Ciencias del Mar y Ambientales, Universidad de Cádiz, Puerto Real, Spain, 5 Canary Islands' Ornithology and Natural History Group (GOHNIC), Buenavista del Norte, Tenerife, Canary Islands, Spain, 6 Departmen of Evolutionary Zoology and Human Biology, University of Debrecen, Debrecen, Hungary, 7 Department of Life Sciences, Imperial College London, Ascot, United Kingdom

* jov23@bath.ac.uk

**Data Availability Statement:** All relevant data are available from GitHub (https://github.com/josevalde/Prevalence-of-bateria-in-plovers/blob/master/dataset%26Rcode.zip).

## Abstract

Do islands harbour less diverse disease communities than mainland? The island biogeography theory predicts more diverse communities on mainland than on islands due to more niches, more diverse habitats and availability of greater range of hosts. We compared bacteria prevalences of *Campylobacter*, *Chlamydia* and *Salmonella* in cloacal samples of a small shorebird, the Kentish plover (*Charadrius alexandrinus*) between two island populations of Macaronesia and two mainland locations in the Iberian Peninsula. Bacteria were found in all populations but, contrary to the expectations, prevalences did not differ between islands and mainland. Females had higher prevalences than males for *Salmonella* and when three bacteria genera were pooled together. Bacteria infection was unrelated to bird's body condition but females from mainland were heavier than males and birds from mainland were heavier than those from islands. Abiotic variables consistent throughout breeding sites, like high salinity that is known to inhibit bacteria growth, could explain the lack of differences in the bacteria prevalence between areas. We argue about the possible drivers and implications of sex differences in bacteria prevalence in Kentish plovers.

## Introduction

Understanding how biological diversity stablishes and evolve has been a tradition in modern ecology [1, 2]. Insights of historic and contemporary research have led ecologists to develop a number of biodiversity theories that are intended to help us predict biodiversity in a given space and/or time. According to the theory of island biogeography, landscape structure shapes species' abundance, where species richness increases as a function of the area sampled [3]. Along a gradient of ecosystems of increasing size, the number of species inhabiting those

**Funding:** JOV was funded by the Comisión Nacional de Investigación Científica y Tecnológica (CONICYT), BECAS CHILE 72170569; TS by a Royal Society Wolfson Merit Award (WM170050) and by the National Research, Development and Innovation Office of Hungary (ÉLVONAL KKP-126949, K-116310); JF and JS by VolkswagenStiftung (Social behavior and diseases: a comparative investigation of island and mainland bird populations). The funders had no role in study design, data collection and analysis, decision to publish, or preparation of the manuscript.

**Competing interests:** The authors have declared that no competing interests exist.

ecosystems will increase rapidly at first, but then the pace slows down for the larger ecosystems [3, but see limitations, 4]. Island biogeography theory has been mainly built upon the study of macroorganisms, with very little consideration towards the biogeography of microorganisms. In fact, whether microbial biogeography should be considered as a discipline has been subject of debate, because it has long been suggested that organisms smaller than 1 mm have a cosmopolitan distribution [5]. However, different studies have documented spatial and temporal structuration of microbial diversity [6, 7]. For example, positive taxa-area relationships have been found in free-living bacteria [8], as well as a reduction in bacterial diversity across islands of decreasing sizes [9].

Symbiotic organisms are in a close and necessary association with other organisms, through either mutualistic, commensal or parasitic associations [10]. This co-dependant interaction adds an important layer of complexity to the island biogeography theory, mainly for including variables from the host that could also be affected by insularity. For example, animals from islands have been proposed to have weaker immune defence, attributed to the founder effect during colonization and to island environments being relatively parasite poor (compared to the mainland) [3, 11, 12]. Examples of the latter include reduced prevalence and diversity of blood parasites and fewer feather lice species in Macaronesian blackcaps (*Sylvia atricapilla*) [13, 14] and reduced viral pathogen diversity and abundance in insular black-spotted pond frogs (*Pelophylax nigromaculatus*) compared to the mainland [15]. However, contrasting results could be found for other host species and microorganisms studied [16].

Studies investigating variation in microorganism prevalence, not microorganism diversity, in relation to area sampled have been considerably less common. Microorganism prevalence, here defined as the percentage of individuals of a population infected with a given microorganism, will depend importantly upon two variables: (i) the ability of the host to defend against the infection, and (ii) on the ability of the microorganism to infect the host. As mentioned before, insularity is thought to, at certain extent, shape immune function because after many generations exposed to low pathogen pressure and diversity, selection favours a reduction in the energy invested in maintaining a robust immune function [17]. However, changes in immune parameters in response to insularity are not as straightforward as initially thought, as Matson et al. [18] found that in bluebirds (*Sialis sialis*) the immune response was stronger in island than in the mainland. Lobato et al. [19] investigated two immune components in bird assemblages from two islands and mainland in Africa, finding that acquired immunity was lower on islands but no differences were seen in the innate immunity. The high microorganism diversity expected in mainland [9] increases the chances of hosts of encountering strains of microorganism of high virulence, that could rapidly spread out across a population and elevate prevalence at a given sampling time [20]. Also, the force of infection (the rate at which susceptible individuals become infected in a population) is expected to increase with population size in the case of having many susceptible hosts present in a large population at any given time, and because in this ecosystem the number of contacts between infected and susceptible individuals is likely to also increment [21–24]. Conversely, in simplified ecosystem (i.e. islands) the reduced pathogen pressure compared to mainland would suggest, in general, lower levels of pathogen prevalence than in the mainland [25–27]. However, to our knowledge just a handful studies have tested aspects of these predictions, and to date there is very little known on how prevalence of bacteria could be affected by insularity.

Here, we tested for the first time the influence of insularity on the transmission patterns of the bacteria *Campylobacter*, *Salmonella* and *Chlamydia* in four populations of Kentish plover (*Charadrius alexandrinus*). These bacteria are known for its importance in wildlife and public health, being usually commensal in poultry but a common cause of gastrointestinal and respiratory disease in wild birds [28, 29]. Kentish plover is a small shorebird, ideally suited for this

purpose because breeds all across Eurasia and Macaronesia (Fig 1), islands where they are year-round residents and represent populations genetically distinctive from the mainland [30]. We compared the cloacal bacteria prevalence of two island populations from Cape Verde and Canary Islands, and two populations breeding in continental Spain (Table 1). Because these

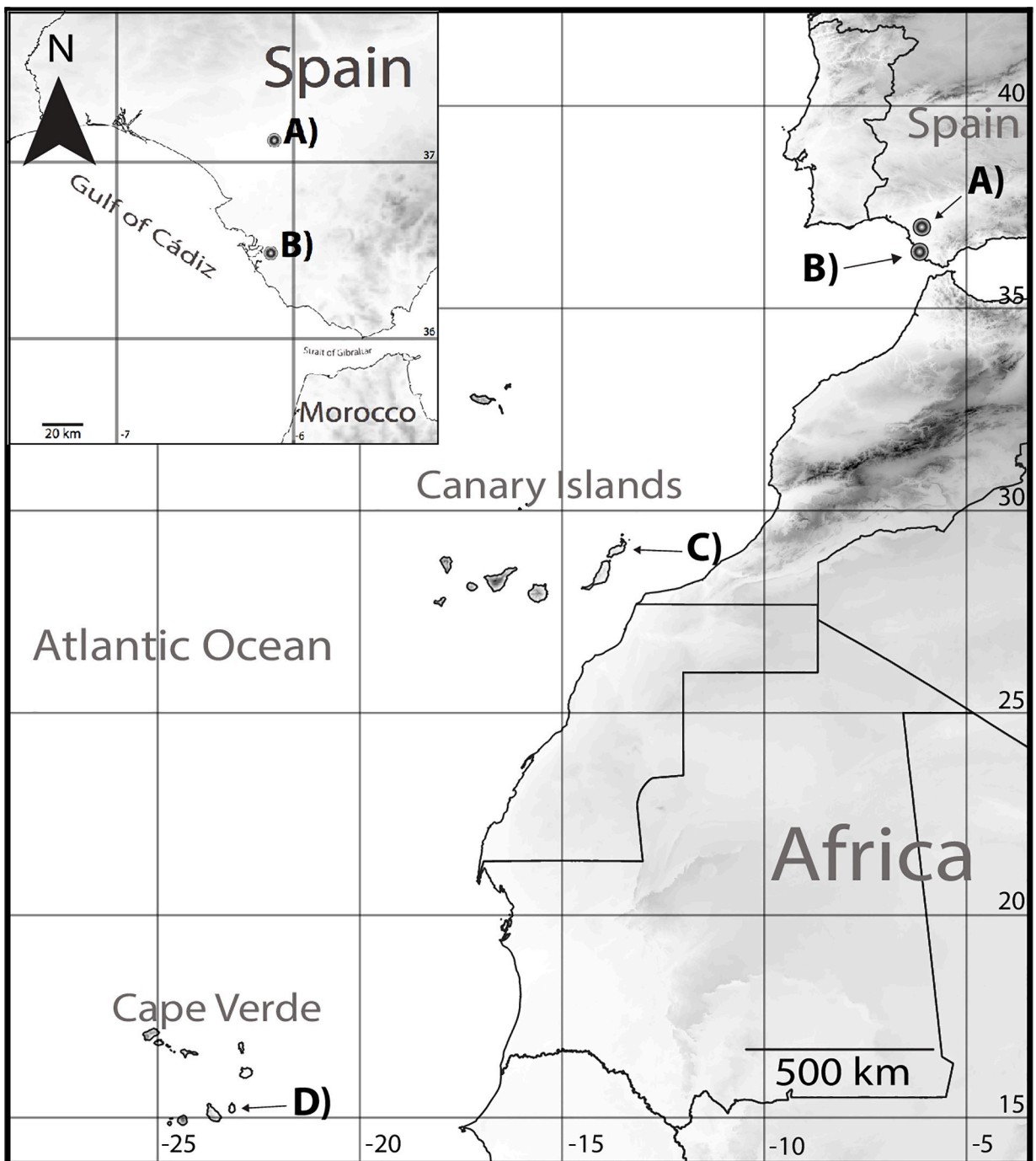

**Fig 1. Sampling locations of the four Kentish plover populations.** Two on mainland and two on islands: (A) Rice fields of Doñana, and (B) Salina la Esperanza, Cádiz, both in Spain; (C) Lanzarote, Canary Islands, and (D) Maio, Cape Verde.

**Table 1. Number of birds examined and infected in mainland and island populations of Kentish plover.**

| | Population | n | Campylobacter | Chlamydia | Salmonella | Pooled infection |
|---|---|---|---|---|---|---|
| Island | Maio, Cape Verde | 88 | 2 | 5 | 10 | 17 |
| | Lanzarote, Canary Islands | 27 | 0 | 0 | 3 | 3 |
| | Total | 115 | 2 | 5 | 13 | 20 |
| Mainland | Doñana, Spain | 52 | 1 | 1 | 10 | 12 |
| | Salina la Esperanza, Cádiz, Spain | 52 | 1 | 4 | 10 | 15 |
| | Total | 104 | 2 | 5 | 20 | 27 |

bacteria could provoke disease, we also investigated the effect of insularity and bacteria infection on body condition. Based on previous evidence, we predicted (i) a higher prevalence of infection in mainland than in insular populations; (ii) bacteria infection will negatively affect the host's body condition [31, 32]; and as possible consequence of the previous two, (iii) birds from islands will have better body condition than those from mainland [33]. Last, because cloacal transmission of bacteria seems to be asymmetrical between the sexes [see 34], we predicted (iv) females potentially having higher prevalences than males.

## Materials and methods

### Ethics

All necessary permits were obtained for the described field studies. Salina la Esperanza: permit granted by the University of Cádiz, Cádiz Bay Natural Park and Animal Health authorities in compliance with Spanish laws (number 2019-/2979/4202/Bc/EA 3619). Doñana: permit granted by the CSIC Ethics committee and Animal Health authorities in compliance with Spanish laws (number 2011_02 21/02/2012/77). Lazarote: permit granted by the Council of Land Usage, Sustainability and Security, Vice-Ministry of Environment, Canary Islands (number 2016/3646). Maio: permit granted by the General Directory of Environment, Cape Verde (number 33/2014).

### Bird species and sampling locations

We captured, ringed, weighed and morphometrically measured breeding Kentish plovers from two different locations in mainland (southern Spain) and from two Macaronesian islands (Table 1, Fig 1). One mainland Kentish plover population was studied in the largest area of rice fields in Spain (36,000 ha) located in a reclaimed marshland behind Doñana National Park, Spain (37°07'08.3"N 6°06'33.7"W). Fieldwork was done in July 2015 at four sites during the peak of the breeding season. The other mainland population bred in a 35-ha saltpan in the Cádiz Bay Natural Park, Puerto Real, Spain (36°30'36.0"N 6°09'20.8"W). Fieldwork was conducted in April–June 2015. Among the island populations investigated, we studied Kentish plover on Lanzarote, Canary Islands (29°03'36.1"N 13°36'24.9"W). Lanzarote is the easternmost island of the archipelago, separated by approx. 120 km from North Africa and 1,000 km from the Iberian Peninsula. Fieldwork was conducted during the breeding season in April–June 2016, monitoring five sites around the island with different environments: saltpans, sandy beaches and semi-desert rocky areas. Lastly, we studied the Kentish plover population in Maio, Cape Verde (15°09'16.6"N 23°11'39.4"W), one of four Sotavento Islands in the archipelago, located at approx. 650 km from West Africa and 2,900 km from the Iberian Peninsula. Approximately 100–200 pairs bred in Maio around areas of saline lakes and saltpans of approx. 100 ha and surrounded by sandy shores. Three sites were monitored during September–November 2015. Kentish plovers present high breeding-site fidelity [35, 36] whereas during

winter the birds from mainland Europe move to SW Europe and W African [37]. Kentish plover from Maio and Lanzarote are year-round residents but eventually could move between islands, particularly in the population of Lanzarote (TS, and GT pers. obs). All field procedures complied with the laws and approved by the ethics committees of the corresponding countries.

### Bacteria sampling and laboratory diagnosis

Our analyses were focused on three bacteria genera of renown importance for wildlife and public health. *Campylobacter* and *Salmonella* are gram-negative bacteria from the Enterobacteriaceae family and often found as commensal microbiota in avian hosts [38, 39]. Commensal strains result from bacterial adaptation to specific hosts [28, 40]. In poultry, many specific strains are recognised as commensal such as *Campylobacter jejuni* ST-104 (ST-21 CC) in broiler [29], but in Kentish plover it is unknown whether there are species-specific strains and to what degree these strains are commensals or can harm host health. Nevertheless, pathogenic strains like *Campylobacter lari* or *Salmonella typhimurium* have been associated with gastrointestinal disease in poultry and in wild birds [29, 30, 41] and are also a latent epidemiological problem causing foodborne disease worldwide [42]. Although these bacteria are typically acquired through an oral-fecal route by ingesting contaminated food or water, evidence shows transmission after copulation, either through direct cloacal contact or due to ingestion of bacteria during post-copulatory preening [43, 34]. *Chlamydia*, on the other hand, are sexually transmitted bacteria and species like *Chlamydia psittaci* may cause chlamydiosis in birds and the zoonosis psittacosis (if transmitted to humans by contaminated aerosols), both being a systemic disease often linked to mortalities [44, 45].

Bacteria sampling took place at capture by gently introducing a sterile cotton swab into the bird's cloaca. Swabs were stored in phosphate-buffered saline (PBS) buffer at -20ºC in the field, and then in the laboratory at -80ºC until further analysis [46]. DNA extraction was done using the Maxwell® 16 Buccal Swab LEV DNA Purification Kit following the manufacturer's protocol. Detection of *Campylobacter* was based on amplification of a DNA segment within the *flaA* short variable region (SVR) of *Campylobacter jejuni* or *C. coli*, according to Ridley et al. [47]. *Chlamydia* detection centred on amplifying the IGS region and domain I of 23S rRNA gene, following Nordentoft et al. [48]. *Salmonella* detection used primers specific for the *invA* gene, as described by Rahn et al. [49]. In brief, real-time PCR assays were conducted with 5 μL of 2 × Rotor-Gene SYBR Green PCR Master Mix, 7 μL of RNase-Free water, 1 μL of primers, and 2 μL of DNA extract. Thermal conditions for PCRs were as follows: initial activation for 10 min at 95ºC, PCR cycling for 15 sec at 95ºC, for 30 sec at 59ºC and for 30 sec at 72ºC (for *Chlamydia*) or 30 sec at 95ºC, for 15 sec at 54ºC and 20 sec at 72ºC (for *Salmonella*) for 45 times, and melting curve were obtained by lowering the temperature from 90ºC to 75ºC, descending by 0.3ºC each step. We used DNA from *Chlamydia psittaci* and *Salmonella typhimurium* as positive controls in each reaction plate. For *Campylobacter* cycling was for 10 sec at 95ºC, for 6 sec at 50ºC and for 6 sec at 72ºC for 35 times, and melting curve were obtained by lowering the temperature from 90ºC to 50ºC, descending by 2.2ºC each step. The positive controls used were *Campylobacter jejuni* and *C. coli*. Negative controls were included in each plate.

### Statistical analyses

Our predictions were tested by running Markov chain Monte Carlo simulations for generalized linear mixed models using the R package 'MCMCglmm' [50]. Differences in insularity were tested by running four models: one for each bacteria type and one for the prevalence of the three bacteria combined (pooled bacteria infection). The models had bacteria infection

(binomial variable: infected/un-infected) as response variable, and insularity (binomial variable: island/mainland) and sex (binomial variable: female/male) as fixed factors as well as the two-way interaction between these two variables. Date of sampling (Julian date) and sampling site were added as random terms. Sampling site corresponded to the sites sampled within each location: four in Doñana, one in Salina La Esperanza, five in Lanzarote, and three in Maio. We used parameter expanded priors for the random effects (list(V = diag(1)*0.02, nu = 7)), and fixed effect priors for binary responses i.e. fixing the residual variance at 1 (list(V = diag(1), nu = 0.002, n = 1, fix = 1)). Models were run across 1,000,000 iterations with thin of 600 and a burn-in of 1,500. These values were determined based on model convergence and autocorrelation levels assessed through the Gelman-Rubin test [51], and trace graphs and the 'autocorr' function, both implemented in the R package 'coda' [52]. In all four models, the potential scale reduction factor was 1.01 or lower, which is below the threshold of 1.1 indicating good model convergence. Autocorrelation was also low, always below the threshold of 0.1 [50].

Body condition was estimated using the scaled mass index proposed by Peig and Green [53], consisting on standardizing body mass for a given size using a body linear measurement (here, wing length). This analysis was conducted only for the pooled bacteria prevalence due to the very low prevalences of *Campylobacter* and *Chlamydia* (see results). This model was run with a Gaussian error distribution and had body condition as response variable and infection status, insularity and sex as fixed factors and their two-way interactions. Date of sampling and sampling site were added as random terms. We used parameter expanded priors for the random effects (same as above) but inverse gamma priors (list(V = 1, nu = 0.002)) for the residuals and normal distributions centred on zero with large variances as fixed effects priors (default prior in MCMCglmm). This model was run across 1,000,000 iterations with thin of 600 and a burn in of 1,500. Here, the potential scale reduction factor was 1.001 or lower and the autocorrelation was not higher than 0.03 [50]. In this analysis 11 Kentish plovers were excluded from the model because wing length, body mass or both measurements were not available. MCMCglmm results are expressed as posterior mean, lower and upper 95% credibility intervals, and significance as a pMCMC value. All statistical analyses were conducted in R v3.3.3 [54].

## Results

Forty-seven out of 219 birds sampled were infected (21.5%). The highest prevalence was recorded in *Salmonella* (15.1%) followed by *Chlamydia* (4.6%) and *Campylobacter* (1.8%), with no birds presenting mixed infections. Bacteria infection was spread out in most populations (Table 1) except on Canary Islands where only *Salmonella* was found (Table 1).

Bacteria prevalence was always higher in birds from mainland than from islands, however this difference was non-significant. Although a trend appeared when infection was pooled together, showing nearly significant higher prevalence of infection in mainland ($P = 0.077$; Fig 2, Table 2). Females had a higher prevalence of *Salmonella* than males, and the same pattern was found when all bacteria were pooled together (Fig 2, Table 2). The interaction between insularity and sex was not significant (all cases $P > 0.05$), so insularity did not affect bacteria prevalences between the sexes.

Body condition was not significantly affected by the presence of the three bacteria (Fig 3, Table 3). Females were heavier than males in the mainland while on islands no sex differences in body condition were found (Table 3).

## Discussion

Our results showed that *Campylobacter*, *Chlamydia* and *Salmonella* were widespread among most Kentish plover populations and similarly prevalent in mainland and islands. Female

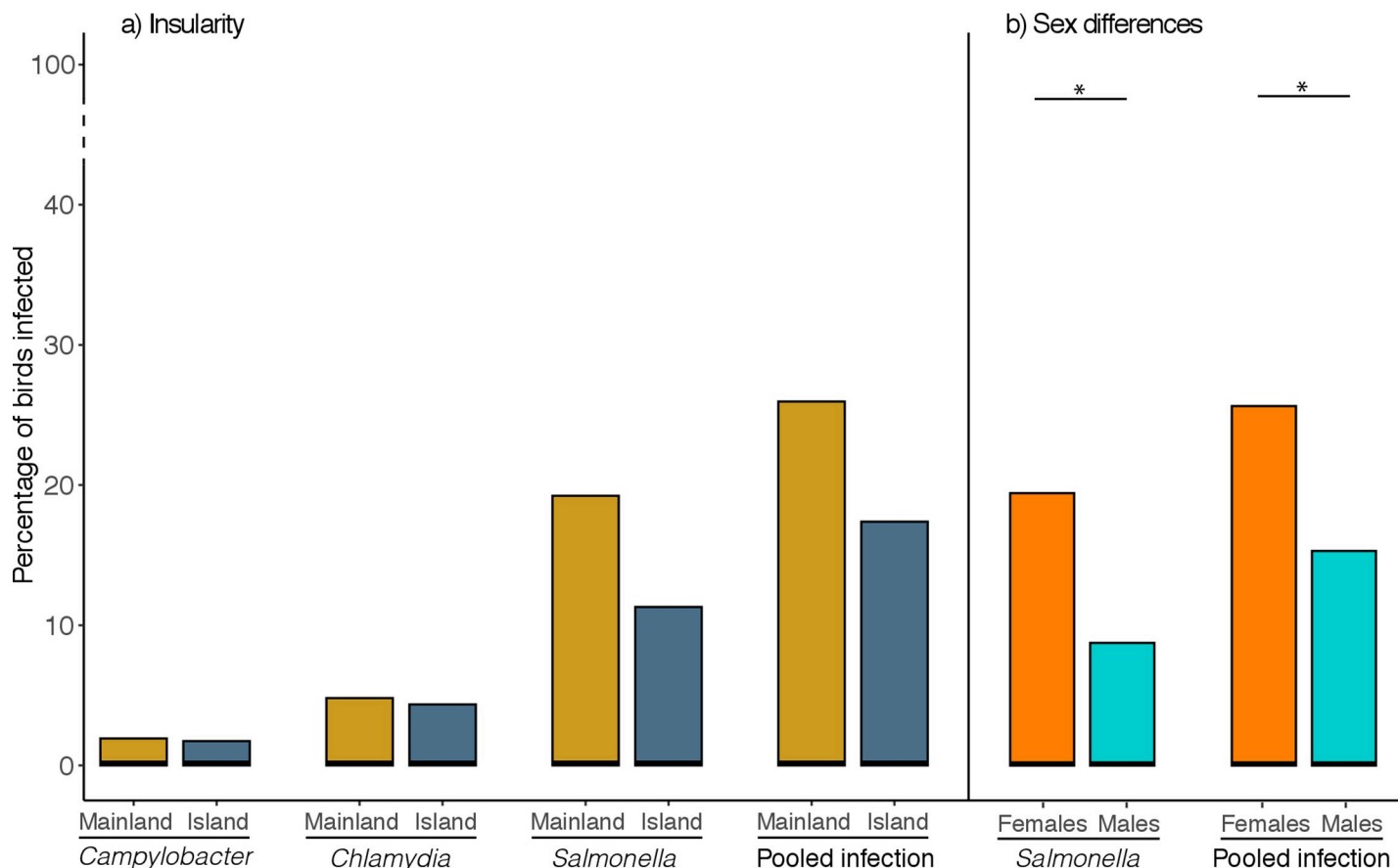

**Fig 2. Differences in bacteria prevalence between populations.** Prevalence of cloacal bacteria infection between a) mainland and island and b) male and female Kentish plovers. *Indicates a statistically significant difference of $P < 0.05$.

Kentish plovers had a higher *Salmonella* prevalence than males, a pattern also found when the infection of the three bacteria was combined together. Lastly, we showed that body condition was not related to infection of the three bacteria but to sex and insularity, with a higher body condition found in females and in birds from the continent.

## Insularity

We found similar *Campylobacter*, *Chlamydia* and *Salmonella* prevalences in insular and mainland bird populations. One reason of our findings in *Salmonella* could originate from the fact that infection with this genus of bacteria, as most shorebird microbiota, depends on the environmental availability of the bacteria [55]. Although Kentish plovers in our study bred in completely different landscapes (i.e. islands vs mainland), the breeding sites were relatively similar in that involved lands of high salinity, scarce vegetation and close to saline water bodies. High levels of salinity could consistently constrain bacteria acquisition throughout sampling locations because salinity is a well-known inhibitor of *Salmonella* and *Campylobacter* growth [56, 57]. Perhaps the exception to this was the population from Doñana that bred near brackish water. Interestingly, the percentage of infected birds in Doñana was equal to those in Cádiz (19.2%) but higher than in Lanzarote and Maio (respectively, 11.1% and 11.4%). The animal diversity in mainland increases the probability of encountering animals hosting *Salmonella* infection that could later be acquired by Kentish plovers [58], and thus is a possible

**Table 2. Infection of (a)** *Campylobacter*, **(b)** *Chlamydia*, **(c)** *Salmonella* **and (d) all bacteria combined in relation to insularity and sex in Kentish plovers (*n* = 219).**

| | | 95% credibility intervals | | |
| --- | --- | --- | --- | --- |
| | Post. mean | Lower | Upper | *P* |
| a) Intercept | -5.196 | -7.807 | -2.984 | **< 0.001** |
| Insularity (island)[a] | 0.002 | -3.833 | 3.726 | 0.995 |
| Sex (males)[b] | 0.533 | -2.906 | 4.687 | 0.754 |
| Insularity (island)[a]*sex (males)[b] | -0.324 | -5.604 | 4.788 | 0.876 |
| Random | | | | |
| Site | 0.027 | 0.006 | 0.063 | |
| Date | 0.028 | 0.007 | 0.066 | |
| b) Intercept | -3.257 | -4.519 | -2.139 | **< 0.001** |
| Insularity (island)[a] | -0.945 | -3.140 | 1.140 | 0.360 |
| Sex (males)[b] | -1.434 | -4.541 | 1.275 | 0.318 |
| Insularity (island)[a]*sex (males)[b] | 2.276 | -1.385 | 6.014 | 0.197 |
| Random | | | | |
| Site | 0.029 | 0.006 | 0.071 | |
| Date | 0.028 | 0.006 | 0.064 | |
| c) Intercept | -1.349 | -1.991 | -0.630 | **< 0.001** |
| Insularity (island)[a] | -0.842 | -1.958 | 0.219 | 0.129 |
| Sex (males)[b] | -1.331 | -2.584 | 0.094 | **0.043** |
| Insularity (island)[a]*sex (males)[b] | 0.520 | -1.272 | 2.644 | 0.578 |
| Random | | | | |
| Site | 0.027 | 0.006 | 0.061 | |
| Date | 0.030 | 0.006 | 0.076 | |
| d) Intercept | -0.887 | -1.53 | -0.230 | **0.008** |
| Insularity (island)[a] | -0.920 | -1.939 | 0.110 | 0.077 |
| Sex (males)[b] | -1.248 | -2.526 | -0.146 | **0.030** |
| Insularity (island)[a]*sex (males)[b] | 0.980 | -0.693 | 2.678 | 0.252 |
| Random | | | | |
| Site | 0.032 | 0.006 | 0.078 | |
| Date | 0.030 | 0.007 | 0.074 | |

Residual variances were fixed at 1. Significant effects in bold.

[a]Relative to mainland.

[b]Relative to females.

reason of the close to significant higher prevalence of *Salmonella* in the continent. However, a possible counter argument is that hot environments (25–35 degrees C) with high relative humidity such as the islands of Maio and Lanzarote, provide suitable conditions for a longer persistence in the environment of the bacteria [59], increasing the potential of between-individuals bacteria transmission through ingestion of bacteria from feathers during preening, posterior to, for example, belly-soaking [more frequent in hot environments, 60] or direct contact of individuals (e.g. copulation) [34]. *Campylobacter* may be also acquired from the environment and thus could be affected by the same factors described for *Salmonella* [61]. However, *Campylobacter* is much more susceptible to environmental conditions, requiring, for example, microaerophilic conditions to proliferate [57, 62]. Another reason of such low prevalences found (1.8%, 4 infected out of 219 birds) could be because direct PCR detection from feaces can be problematic compared to enrichment and culture, regarded as the gold

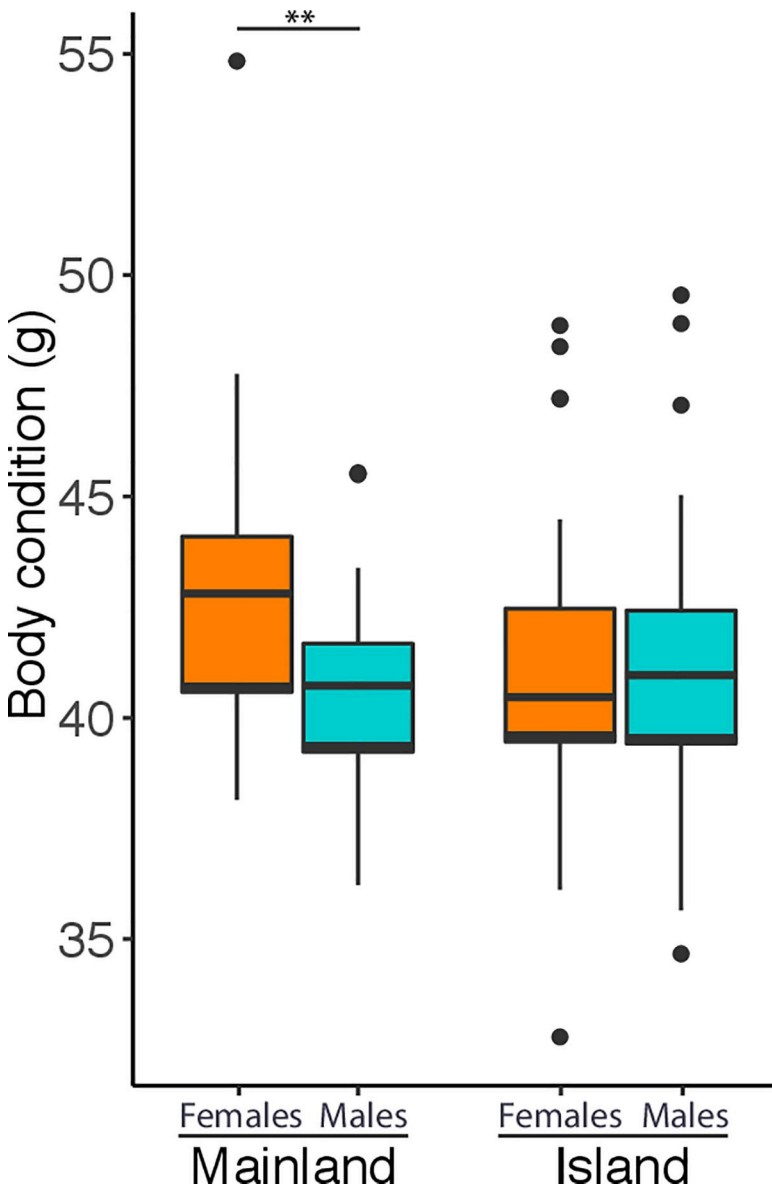

**Fig 3. Variation in body condition in Kentish plovers.** Scaled mass index in male and female Kentish plovers from mainland and islands (females and males from mainland weighed on average [mean ± standard deviation] 42.6 ± 2.9 and 40.5 ± 2.1 g, respectively, while on islands, females and males weighed 41.0 ± 2.8 and 41.0 ± 3.1 g, respectively). Medians, upper and lower quartiles are shown. Whiskers indicate minimum and maximum values and circles outliers. **Indicates a statistically significant difference of $P < 0.01$.

standard for *Campylobacter* detection [63]. Although *Campylobacter* prevalences in the wild are medium to high and around 75% in shorebirds [64], a previous study investigating the prevalence of *Campylobacter* spp. in Kentish plovers failed to find any infected individual out of 12 tested [65]. In addition, low prevalences, as found in *Campylobacter* and *Chlamydia*, may make more difficult to detect differences in prevalence between populations. This is particularly important for bacteria like *Chlamydia*, that is mainly horizontally transmitted by direct contact between infected individuals [66].

**Table 3. Factors affecting the body condition in Kentish plovers (*n* = 208).**

| | | 95% credibility intervals | | |
|---|---|---|---|---|
| | **Post. mean** | **Lower** | **Upper** | ***P*** |
| **Intercept** | **42.656** | **41.802** | **43.486** | **< 0.001** |
| Pooled bacteria infection | -0.195 | -1.578 | 1.151 | 0.789 |
| Insularity (island)[a] | -1.815 | -2.964 | -0.639 | **0.001** |
| Sex (males)[b] | -2.196 | -3.445 | -1.029 | **0.001** |
| Pooled bacteria infection*Insularity (island)[a] | 0.958 | -0.888 | 3.081 | 0.342 |
| Pooled bacteria infection*sex (males)[b] | 0.198 | -1.730 | 2.407 | 0.841 |
| Insularity (island)[a]*sex (males)[b] | 2.236 | 0.702 | 3.838 | **0.006** |
| Random | | | | |
| Site | 0.028 | 0.006 | 0.071 | |
| Date | 0.029 | 0.006 | 0.067 | |
| Residual | 8.047 | 6.529 | 9.587 | |

Eleven birds were excluded from the model. Significant effects in bold.

[a]Relative to mainland.

[b]Relative to females.

## Sex-specific bacteria prevalence

Bacteria prevalence was significantly female-biased when *Salmonella* infection and all the bacteria species were analysed together. These sex differences were independent of insularity. In addition to potential differences in the ecology of the different bacteria genera, the very low prevalences of *Chlamydia* and *Campylobacter* may explain why our results only approached significance for *Salmonella*. Studies of sex-specific parasite infection (as general term) have shown great heterogeneity in their patterns and are rather scarce in terms of bacteria presence. One study investigating pathogen prevalence in the island populations of Berthelot's pipit (*Anthus berthelotii*) found no sex differences of infection with pox virus, *Plasmodium* and *Leucocytozoon* [67]. Another study found that females had higher prevalence of cloacal bacteria than males in alpine accentor (*Prunella collaris*) [68], while bacteria richness did not vary with sex in blue tits (*Cyanistes caeruleus*) [69]. This heterogeneity may be due to many non-exclusive factors, including differences in immunocompetence and behavior between host sexes as well as differences in the ways of transmission between the pathogens studied [70–72]. The immune system plays an important role in pathogen defense hence if sex-specific differences in immunocompetence exist, we could expect unbalanced infection. However, although a recent meta-analysis showed in general no sex differences in immune capacity across animals [including birds, 73], the literature available shows plenty of variation (i.e. female and male biases) at species and population level that has not yet been explained [74–77].

## Body condition

Recent advances in methods of microorganism detection have shown that wild animals often are natural reservoir of pathogenic microorganisms without any apparent health cost [38, 78, 79]. The relationship between bacteria infection and body condition could be difficult to untangle because one could argue that individuals in poor body condition would be more prone to infection, however, this is more likely to happen when access to food is reduced and immunity also gets compromised [80, 81]. Nevertheless, the consistent presence of these bacteria that we found throughout the locations and populations of Kentish plover, in addition to the lack of impact on body condition, suggests that these shorebirds are natural reservoir of

*Campylobacter*, *Chlamydia* and *Salmonella*. However, these bacteria have great diversity of strains with different pathogenicity and the impact of parasites on host health, survival and life history is difficult to demonstrate based on observational studies and always require experimental manipulation of parasite prevalence or intensity of infection [82, 83]. Further studies are needed to determine whether positive birds harboured strains distinctive and specific to Kentish plover, or strains of other species that have recently adapted to this host [28]. Contrary to our expectations, birds from mainland had better body condition than those from islands. Animals living on islands are exposed to low interspecific competition for food [84]. In addition, the tropics lack of well-defined seasons, with rather stable temperatures during the day and night, and predictable foraging conditions. Such conditions could prevent birds from fuelling up excessively and store energy as fat because of the constant food availability. Also, animal in tropics tend to have slower basal metabolic rates, which imply lower caloric requirements [85]. On the other hand, birds from mainland are exposed to more variable environmental condition like lower temperatures at night, that might translate into higher food consumption during the day [86]. Interestingly, only birds from mainland had sex differences in body condition. Sex differences in body mass or body condition during breeding usually occur previous to the egg-laying stage, where females increase their body mass, and then later during the nesting and brood care stage, where the sex that provides most of the care will see the detrimental effects on body weight [e.g. 87–89]. If we take this into consideration, in addition to previous studies in different continental Kentish plover populations describing a polyandrous mating system [90], we could argue that a polyandrous mating system could explain why only in mainland we found that females were heavier than males, because in polyandry males provide the brood case. However, this postulate cannot be confirmed because to date, no empiric studies have investigated the mating system of the mainland populations here studied.

## Conclusion

Although in a relatively low prevalence, our study shows that *Campylobacter*, *Chlamydia* and *Salmonella* were widely present across Kentish plover populations, placing it as possible natural reservoirs of these bacteria. Contrary to our expectations, the three bacteria examined were equally prevalent on mainland and on island populations. Insularity and the sex of the host were important variables determining the bird's body condition, but these patterns were difficult to interpret. Positive relationships between geographical size and animal, plant and bacteria diversity have been reported [e.g. 9, 91]. In the case reported here, it is possible that bacteria infection in hosts do not directly depend on geographical size because of the added level of complexity of including the many variables of the host that could also be affected by insularity. We emphasize on expanding research on bacteria infection in wild birds from an ecological point of view, necessary to further understand the potential impact of social interactions and mating system structure on sexual differences in the prevalence of cloacal bacteria.

## Acknowledgments

We thank Francisco Miranda for conducting the lab work, Mabel Mena for her help in georeferencing systems, Alberto Pastoriza, Carlos Moreno and Manuel Vázquez for their contribution during fieldwork in Doñana, and Carlos Armas, Jaime Camacho and Ico Tejera for their help in the fieldwork in Lanzarote. We also thank the Spanish Regional Governments and the Environment General Office (DGA) of Cape Verde for kindly authorizing fieldwork.

## Author Contributions

**Conceptualization:** Josué Martínez-de la Puente, Tamás Székely, Julia Schroeder, Jordi Figuerola.

**Data curation:** Josué Martínez-de la Puente, Jordi Figuerola.

**Formal analysis:** José O. Valdebenito.

**Funding acquisition:** Tamás Székely, Julia Schroeder, Jordi Figuerola.

**Methodology:** Jordi Figuerola.

**Resources:** Macarena Castro, Alejandro Pérez-Hurtado, Gustavo Tejera.

**Supervision:** Josué Martínez-de la Puente, Jordi Figuerola.

**Visualization:** José O. Valdebenito, Naerhulan Halimubieke.

**Writing – original draft:** José O. Valdebenito.

**Writing – review & editing:** José O. Valdebenito, Josué Martínez-de la Puente, Macarena Castro, Alejandro Pérez-Hurtado, Gustavo Tejera, Tamás Székely, Naerhulan Halimubieke, Julia Schroeder, Jordi Figuerola.

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
