## [Decision Letter · Decision Letter 0]

10 Jul 2020

PONE-D-20-15756

Association of insularity to cloacal bacteria prevalence in a small shorebird

PLOS ONE

Dear Dr. Valdebenito,

Thank you for submitting your manuscript to PLOS ONE. After careful consideration, we feel that it has merit for PLOS ONE’s publication after minor corrections. Therefore, we invite you to submit a revised version of the manuscript that addresses the points raised during the review process.

I would like to apologize for the delay in the response, but it was very difficult to find reviewers. A total of 19 researchers were invited to review the manuscript, and only one of them accepted. Below you will find this review and also the review that I performed.

We look forward to receiving your revised manuscript.

Kind regards,

Magdalena Ruiz-Rodriguez

Academic Editor

PLOS ONE

Editor Review:

This is a very nice and interesting work with a large dataset, which allows to obtain robust results that are well explained and discussed.

I have some comments on the manuscript:

Title: there is a part of the manuscript related to body condition apart from bacteria (e.g. L320-339). Therefore, I suggest to include this in the title, for example: “…prevalence and to body condition in a…”

L99: The first prediction has been properly explained before. However, the other three predictions are not previously introduced. For example, we do not know why we could expect differences between sexes (prediction ii), and although it is intuitive, there are no arguments in the Introduction to expect the negative relationship between bacteria and body condition (prediction iii). Finally, concerning to prediction iv, the relationship between the immunity and habitat (island vs. mainland) has been explained, but this is not the case with the body condition. Although the immunity is part of the general condition, here you refer specifically to the scale mass index, which could be positively or negatively related with the immune response.

Therefore, before list the predictions, all of them should be properly introduced.

L136: Change sampled to sampling.

L189 and Fig. 3: In the text, authors say that they used as body condition the scale mass index. However, in the Fig. 3, it seems like if you used the body mass, and actually, it is expressed in grams (g). Please, clarify this point.

L251: Change to “reason”

L351: change to “bacterial infection”

2. In your Methods section, please provide additional location information of the study sites, including geographic coordinates for the data set if available.

4. We note that Figure 1 in your submission contain map images which may be copyrighted. All PLOS content is published under the Creative Commons Attribution License (CC BY 4.0), which means that the manuscript, images, and Supporting Information files will be freely available online, and any third party is permitted to access, download, copy, distribute, and use these materials in any way, even commercially, with proper attribution. For these reasons, we cannot publish previously copyrighted maps or satellite images created using proprietary data, such as Google software (Google Maps, Street View, and Earth). For more information, see our copyright guidelines: http://journals.plos.org/plosone/s/licenses-and-copyright.

4.1.    You may seek permission from the original copyright holder of Figure 1 to publish the content specifically under the CC BY 4.0 license.

4.2.    If you are unable to obtain permission from the original copyright holder to publish these figures under the CC BY 4.0 license or if the copyright holder’s requirements are incompatible with the CC BY 4.0 license, please either i) remove the figure or ii) supply a replacement figure that complies with the CC BY 4.0 license. Please check copyright information on all replacement figures and update the figure caption with source information. If applicable, please specify in the figure caption text when a figure is similar but not identical to the original image and is therefore for illustrative purposes only.

Reviewers' comments:

Reviewer's Responses to Questions

**Comments to the Author**

1. Is the manuscript technically sound, and do the data support the conclusions?

Reviewer #1: Yes

2. Has the statistical analysis been performed appropriately and rigorously? 

Reviewer #1: Yes

3. Have the authors made all data underlying the findings in their manuscript fully available?

Reviewer #1: Yes

4. Is the manuscript presented in an intelligible fashion and written in standard English?

Reviewer #1: Yes

5. Review Comments to the Author

Reviewer #1: Dear Editor and authors. This is a well-written short paper with an interesting point of view and decent sample size. I like the idea of looking at specific bacterial prevalence as a variable to test between mainland and island populations.

My main question while reading this paper regards as to why the authors choose throughout the paper present these bacterial genera as established pathogens? The authors seem to imply it is presumed these organism cause harm and disease to the host. However, it is well known that these bacterial genera may often present no adverse host effects in birds. This has been extensively studied in chickens where they are often found as commensals. The authors seem aware of this fact, since they write in line 38: “Campylobacter and Salmonella are gram-negative bacteria from the Enterobacteriaceae family and often found as commensal microbiota in avian hosts [36, 37].” Seeing how the authors may pick up different species from the studied genera, how are they able to tell that the obtained organisms are in fact pathogenic in these birds?

Further minor discussion points

• Seeing how these bacteria are often acquired through sexual transmission (line 144), is it not of importance to account for host age? One would assume younger individuals have experienced fewer sexual encounters and less possibilities for transmission. I understand it may be difficult to assess host age in these birds, but even so it might deserve a mention somewhere.

• L147 brings up Chlamydia psittaci, but there’s no evidence in the current study that this is the bacterial species that is being sampled.

• I’m missing some details in the Methods how the three bacterial genera were amplified. I’m aware the authors refer to other studies, but it would be good if they can add some further details so a reader does not have to dig up and read through an additional three papers just to figure out what the authors sequenced.

• L258: “salinity is a well-known inhibitor of bacteria growth” Yes, but what is the effect of salinity on these three specific genera that is sampled? That’s the important point to discuss.

• Question: Is higher body mass always regarded as having better body condition?

6. PLOS authors have the option to publish the peer review history of their article (what does this mean?). If published, this will include your full peer review and any attached files.

Reviewer #1: No

---

## [Author Response · Author response to Decision Letter 0]

20 Jul 2020

All comments have been addressed in the file labelled Response to Reviewers.

---

## [Decision Letter · Decision Letter 1]

24 Jul 2020

Association of insularity and body condition to cloacal bacteria prevalence in a small shorebird

PONE-D-20-15756R1

Dear Dr. Valdebenito,

We’re pleased to inform you that your manuscript has been judged scientifically suitable for publication and will be formally accepted for publication once it meets all outstanding technical requirements.

Kind regards,

Magdalena Ruiz-Rodriguez

Academic Editor

PLOS ONE

Additional Editor Comments:

I think that the manuscript has substantially improved and it is almost ready to be published. I have only one comment. At the end of Introduction (L106-108), the last prediction is not very clear. The transmission of cloacal bacteria among sexes may occur in both directions; here you refer to a reference (35: 35. Sandercock BK, Székely T, Kosztolányi A. The effects of age and sex on the apparent survival of Kentish plovers breeding in southern Turkey. Condor. 2005; 107(3):583–596. doi: 10.1650/0010-5422(2005)107[0583:TEOAAS]2.0.CO;2) in which it is not supported your statement of females being negatively affected by cloacal transmission. If you mean that females may have more bacterial load than males due to cloacal transmission, please change the reference and clarify the sentence.

Reviewers' comments:

**Comments to the Author**

1. If the authors have adequately addressed your comments raised in a previous round of review and you feel that this manuscript is now acceptable for publication, you may indicate that here to bypass the “Comments to the Author” section, enter your conflict of interest statement in the “Confidential to Editor” section, and submit your "Accept" recommendation.

Reviewer #1: All comments have been addressed

2. Is the manuscript technically sound, and do the data support the conclusions?

Reviewer #1: (No Response)

3. Has the statistical analysis been performed appropriately and rigorously? 

Reviewer #1: (No Response)

4. Have the authors made all data underlying the findings in their manuscript fully available?

Reviewer #1: (No Response)

5. Is the manuscript presented in an intelligible fashion and written in standard English?

Reviewer #1: (No Response)

6. Review Comments to the Author

Reviewer #1: I believe the authors have answered my questions and adequately revised the manuscript.

The only minor comment I have is that the dryad link the authors provided does not work:

https://doi.org/10.5061/dryad.9ghx3fffg

7. PLOS authors have the option to publish the peer review history of their article (what does this mean?). If published, this will include your full peer review and any attached files.

Reviewer #1: No

---

## [Editor Report · Acceptance letter]

7 Aug 2020

PONE-D-20-15756R1 

Association of insularity and body condition to cloacal bacteria prevalence in a small shorebird 

Dear Dr. Valdebenito:

I'm pleased to inform you that your manuscript has been deemed suitable for publication in PLOS ONE. Congratulations! Your manuscript is now with our production department. 

Kind regards, 

on behalf of

Dr. Magdalena Ruiz-Rodriguez 

Academic Editor

PLOS ONE